# Reduced Graphene Oxide-Coated CuFeO_2_ with Fenton-like Catalytic Degradation Performance for Terramycin

**DOI:** 10.3390/nano12244391

**Published:** 2022-12-09

**Authors:** Liping Wang, Gonghao Liu, Mingyu Zhang, Kun Luo, Ya Pang

**Affiliations:** 1College of Materials and Environmental Engineering, Changsha University, Changsha 410022, China; 2State Key Laboratory for Powder Metallurgy, Central South University, Changsha 410083, China

**Keywords:** rGO-coated CuFeO_2_, hydrothermal reaction, fenton-like, terramycin

## Abstract

A novel Fenton-like catalyst made of reduced graphene oxide-coated CuFeO_2_ (rGO-coated CuFeO_2_) was synthesized by the hydrothermal reaction method to remove terramycin from aqueous solutions. The catalytic degradation performance of rGO-coated CuFeO_2_ for terramycin was verified with H_2_O_2_ activation. The characterization reveals that rGO-coated CuFeO_2_ has a micro- and mesoporous structure, with groups such as C=C/C−C, CH_2_−CO, and HO−C=O found on the surface. The Fenton-like catalytic degradation of terramycin by rGO-coated CuFeO_2_ was in line with the pseudo-second-order kinetic model, and the elevated temperature accelerated the reaction. Terramycin was catalytically degraded by rGO-coated CuFeO_2_ in two steps: terramycin was first adsorbed by rGO, and then Fenton-like degradation took place on its surface. This research presents new insight into the design and fabrication of Fenton-like catalysts with enhanced performance.

## 1. Introduction

Antibiotics are widely used in human and livestock farming as the main drugs to treat infections [1]. However, antibiotics are difficult to fully degrade in humans and animals, and the residual antibiotics lead to widespread drug-resistant genes [2,3], and the consequences are unimaginable [4]. Among them, tetracycline antibiotics such as terramycin are mainly used in livestock and poultry breeding; these antibiotics are difficult to biodegrade and highly toxic, and traditional water treatment technology cannot effectively remove them. Therefore, it is imperative to develop new technology for the efficient removal of these antibiotics from aqueous solutions.

At present, for the treatment of antibiotic wastewater, advanced oxidation technology has many advantages compared with other treatment technologies, while the Fenton process is one of the most significant advanced oxidation processes because it attacks antibiotics with generated reactive oxygen species and has exhibited great potential for such water decontamination. The Fenton process is currently the most efficient and thorough technology available to remove antibiotics from aqueous solutions due to its powerful oxidation capacity and environmental friendliness [5,6,7,8]. However, the traditional Fenton process will result in a loss of catalytically active ions and poor catalytic efficiency. While Fenton-like technology, which compensates for the shortcomings of traditional Fenton technology, has become the focus of recent research [9,10,11,12,13,14].

The most important part of the Fenton-like technology is the construction of Fenton-like catalysts. Despite the fact that there have been numerous studies on Fenton-like catalysts for the degradation of antibiotics, these catalysts are typically based on iron, such as nanoscale zerovalent iron [10,15], Fe_2_O_3_ [16], Fe_3_O_4_ [17], and FeOOH [18], but they have disadvantages like poor stability and low catalytic performance. Regulation and optimization of the surface structure of catalysts is a good way to give rise to active species in Fenton-like processes. Recently, researchers have found that bimetallic compounds have Fenton-like catalytic synergies [19,20,21,22]. Liu et al. [19] used an Al^0^-CNTs-Cu_2_O composite to degrade sulfamerazine by activating O_2_ to generate reactive oxygen species (ROS). In another study [20], magnetic core-shell MnFe_2_O_4_@C and MnFe_2_O_4_@C-NH_2_ nanoparticles showed that the COD removal efficiency after 180 min was 63.8% and the corresponding BOD_5_/COD increased from 0.012 to 0.36. Furthermore, some studies have shown that Cu-Fe bimetallic oxides have efficient synergistic effects to activate hydroxyl radicals produced by hydrogen peroxide [21,22,23,24]. In particular, CuFeO_2_ was found to have high catalytic performance and no toxicity. However, CuFeO_2_ agglomerates easily, which reduces both the specific surface area and the catalytic activity.

On the other hand, graphene has a large specific surface area and good electron transport ability, and it is easy to adsorb organic compounds with an aromatic ring. Therefore, graphene compounds have attracted extensive attention in the field of Fenton-like processes to remove antibiotics [25,26,27,28,29,30,31]. However, few studies have explored the Fenton-like catalytic degradation of antibiotics by a graphene and CuFeO_2_ combination.

Therefore, in this research, using CuCl_2_ and Fe(NO_3_)_3_•9H_2_O as precursors, a novel Fenton-like catalyst made of rGO-coated CuFeO_2_ was prepared by the hydrothermal reaction process, combining the advantages of graphene and CuFeO_2_. The prepared rGO-coated CuFeO_2_ was characterized, and systematically, its catalytic performance by activating H_2_O_2_ for the degradation of terramycin was evaluated. The Fenton-like catalytic degradation pathways of rGO-coated CuFeO_2_ for terramycin in an aqueous solution were revealed by performing electron paramagnetic resonance (EPR). Based on the results, a good process mechanism for terramycin removal was proposed.

## 2. Materials and Methods

### 2.1. Chemicals and Materials

All chemical reagents were used as received without further purification. Graphene oxide solution (TNWGO-3) was obtained from the Zhongke Timesnano Company, Chengdu, China. Thirty percent hydrogen peroxide and 36~38% HCl were used in our experiment. Other chemicals used in this study were of analytical-laboratory grade. In addition, 0.1 M NaOH and 1M HCl were prepared to adjust the pH values of the solutions.

### 2.2. Catalyst Preparation

An amount of 2.025 g CuCl_2_ and 6.06 g Fe(NO_3_)_3_•9H_2_O were dissolved in 60 mL ultrapure water, then 10 mL GO solution (5 mg/L) and 5.0 g NaOH were added in order and ultrasonically stirred at 25 °C for 1 h. After that, the above-mixed solution was poured into a Teflon-lined stainless-steel autoclave and hydrothermally reacted at 180 °C for 12 h. Next, the black precipitates were obtained and washed with ultrapure water and ethanol to neutralize, then put into the oven and dried at 80 °C for 4 h. Finally, the material was ground into powder. Its formation mechanism is shown in Figure 1.

### 2.3. Characterizations

The scanning electron microscope (Zeiss Sigma 300, Oberkochen, Germany) combined with an energy-dispersive X-ray spectrometer (Smartedx, Oberkochen, Germany) was used to observe the microstructure. Transmission electron microscopy (TEM, JEOL JEM-2100, Tokyo, Japan) was employed to further investigate the structure and obtain detailed information about rGO-coated CuFeO_2_. The surface chemistry of rGO-coated CuFeO_2_ was analyzed by Fourier transform infrared spectroscopy (Thermo Scientific Nicolet iS5, Waltham, MA, USA). The Brunauer–Emmett–Teller (BET) specific surface area and porous size distribution were identified using the N_2_ adsorption-desorption test at 77 K (Micromeritics Instrument ASAP 2020HD88, Atlanta, GA, USA). The element’s chemical states in rGO-coated CuFeO_2_ were investigated using the X-ray photoelectron spectrometer (XPS, Thermo Scientific K-Alpha^+^, Waltham, MA, USA). The crystal structures of rGO-coated CuFeO_2_ were identified by the powder X-ray diffraction (XRD) measurement. On a D8 Advance diffractometer (Bruker, Karlsruhe, Germany) equipped with Cu-Kα radiation (30 kV and 40 mA), the XRD patterns were recorded in the 2θ range of 5–90° at a scanning rate of 5° min^−1^. The electron paramagnetic resonance (EPR) analysis was carried out on a Bruker A300 spectrometer (Bruker, Karlsruhe, Germany).

### 2.4. Batch Experiments

The batch experiments were conducted in conical flasks using a thermostat oscillator in a water bath. At first, 100 mL of terramycin solution with a concentration of 35 mg/L was poured into a conical flask, and then 0.1 g of rGO-coated CuFeO_2_ was added to the conical flask. Thereafter, H_2_O_2_ was added until the concentration of H_2_O_2_ reached 140 mmol/L, and the pH was adjusted to 5 with 0.1 M HCl. The conical flasks were oscillated at 110 r/min for a time under 25 °C, 35 °C, and 45 °C, respectively. The concentrations of terramycin were detected using an A360 UV-visible spectrophotometer.

## 3. Results and Discussion

### 3.1. Structure and Characterization

#### 3.1.1. Morphology and the Content Distribution of Elements

SEM and EDX are often used to observe the microscopic morphology of materials and analyze the microregion composition of materials. The morphology of rGO-coated CuFeO_2_ was observed by SEM in Figure 2a,b, which has many particles with crystal shapes that are stacked. The EDX spectra of rGO-coated CuFeO_2_ linked to SEM are seen in Figure 2c, and the element’s content measured by the EDX spectra is listed in Table 1, which showed that Cu atom and Fe atom were at approximately the same proportion of 1:1 and matched with the stoichiometry of the chemical formula (CuFeO_2_). Figure 2d is the real-time surface distribution map, and the elemental mapping images are found in Figure 2e, indicating the coexistence and uniform distribution of C, O, Fe, and Cu elements in rGO-coated CuFeO_2_.

#### 3.1.2. Surface Structure and Properties

XRD is usually used to determine the crystal structure and phase of the substance using the diffraction principle. The crystallographic structure of rGO-coated CuFeO_2_ was confirmed by XRD patterns, as illustrated in Figure 3a. The characteristic peaks of rGO-coated CuFeO_2_ were identified as those of CuFeO_2_ (JCPDS 39-0246), declaring the material containing pure-phase CuFeO_2_ successfully prepared. The TEM observation shown in Figure 3b,c visibly shows that rGO-coated CuFeO_2_ belongs to a core-shell structure with a size of about 300 nm, and CuFeO_2_ is encapsulated by rGO. In addition, it is seen from the HRTEM image in Figure 3d that the crystal spacing of 0.287 and 0.225 nm corresponds to planes 006 and 104 of CuFeO_2_, respectively, implying CuFeO_2_ is formed.

In order to understand the specific surface area and pore structure of rGO-coated CuFeO_2_, nitrogen adsorption and desorption experiments were carried out. It can be seen from Figure 4a that the nitrogen adsorption/desorption isotherms of rGO-coated CuFeO_2_ exhibited the III type with an H3 hysteresis loop, manifesting that the adsorbent-adsorbate interaction is weak. The H3 hysteresis loop is observed with aggregates of plate-like particles giving rise to slit-shaped pores [32], which is consistent with the morphology of rGO-coated CuFeO_2_. The BET specific surface area is calculated as 4.295 m^2^/g using the N_2_ adsorption-desorption isotherms. The DFT pore diameter distribution curve is shown in Figure 4b, making it clear that the pores caused by particle accumulation are mainly micropores and mesopores.

To further investigate the chemical composition and surface characteristics of the material, rGO-coated CuFeO_2_ was analyzed by XPS, and the distinct peaks allocated to elements C, Fe, Cu, and O were discovered in Figure 5a. C 1s spectra were illustrated in Figure 5b, and the three peaks at 284.8, 285.5, and 288.7 eV corresponding to C=C/C−C, CH_2_−CO, and HO−C=O were found. The XPS spectra of Fe 2p for rGO-coated CuFeO_2_ are shown in Figure 5c, and two peaks of Fe 2p_3/2_ and Fe 2p_1/2_ individually locate at 711.4 and 724.8 eV, with a satellite peak appearing at 719.5 eV. It was suggested that most Fe is in a chemical state of three valences on the surface of rGO-coated CuFeO_2_ [31]. In short, the peaks at 711.2 and 724.6 eV were assigned to Fe^3+^ species, while the peaks at higher BEs of Fe 2p_3/2_ and Fe 2p_1/2_ corresponded to Fe^2+^ [22]. Similarly, the Cu 2p high-resolution XPS spectrum is displayed in Figure 5d, and the peaks of binding energy at 932.7 and 952.5 eV are attributed to Cu 2p_3/2_ and Cu 2p_1/2_. The narrower peaks of Cu 2p_3/2_ and Cu 2p_1/2_ are considered Cu^+^ species. The other two peaks belong to the Cu^2+^ species. Figure 5e shows the high-resolution spectrum of O 1s, which can be fit into three peaks at 533.1, 531.7, and 530.1 eV, respectively, corresponding to the oxygen of physically absorbed H_2_O molecules (H_2_O)/C=O [33], oxygen in the hydroxyl group (−OH), and lattice oxygen (O^2−^) in the delafossite structure.

In addition, FTIR is commonly used to analyze the surface chemical properties of materials. To better understand the chemical properties of the material surface, FTIR was carried out and shown in Figure 5f. In Figure 5f, the absorption peak at 3410 cm^−1^ is caused by the stretching vibration of −OH, the absorption peak at 1576 cm^−1^ is attributed to the stretching vibration of the aromatic ring C=C, and the absorption peak at 1224 cm^−1^ is due to the stretching vibration of C−O, which are consistent with the results measured by XPS. Additionally, the absorption peak at 659 cm^−1^ corresponds to the stretching vibration of C−Cl formed because of the hydrothermal reaction. In addition, the absence of the C=O peak in HO−C=O may be due to hydrogen bonding between C=O and −OH, which reduces the vibration frequency of C=O.

### 3.2. Catalytic Performance of rGO-Coated CuFeO_2_ in a Fenton-like Process

The catalytic performance of the rGO-coated CuFeO_2_ was evaluated by activating H_2_O_2_ for terramycin degradation. Figure 6a shows the results of terramycin removal in the rGO-coated CuFeO_2_/H_2_O_2_ system at varying temperatures. It can be seen that the decontamination performance of the rGO-coated CuFeO_2_/H_2_O_2_ system improved with rising temperatures. The *C_t_*/*C*_0_ is 0.653 at 25 °C and decreases to 0.457 at 45 °C when the catalytic degradation time is 250 min, indicating that the reaction is more intense and complete at higher temperatures.

To further understand the speed of catalytic degradation, pseudo-first-order kinetics and pseudo-second-order kinetics were applied to fit the dynamic process of Fenton-like reaction [34]. The pseudo-first-order kinetics and pseudo-second-order kinetics were expressed in Formulas (1) and (2):(1)$$\ln\left(C_{t}/C_{0}\right)=-k_{1}t,$$
(2)$$\frac{1}{C_{t}}-\frac{1}{C_{0}}=k_{2}t,$$
in the formulas, *C_t_* and *C*_0_ correspond to the concentration of terramycin at time *t* and the initial solution, respectively (mg/L), and *k*_1_ and *k*_2_ represent the degradation rate constants of the pseudo-first-order kinetics (min^−1^) and the pseudo-second-order kinetics (L·mg^−1^·min^−1^).

The dynamic fitting curves are shown in Figure 6b,c, and the results of the kinetic parameters obtained from the fittings are summarized in Table 2. Based on the correlation coefficients, the catalytic degradation kinetics showed good compliance with the pseudo-second-order model, as already observed for other Fenton-like catalysts [34]. Table 2 displays that when the temperatures are 25 °C, 35 °C, and 45 °C, the degradation rate constants are 7.907 × 10^−5^ min^−1^, 1.018 × 10^−4^ min^−1^, and 1.467 × 10^−4^ min^−1^, respectively, indicating that the catalytic degradation reaction is accelerated with the increase in reaction temperature. It may be because the increase in reaction temperature accelerates the activation of the rGO-coated CuFeO_2_ for H_2_O_2_ and thus the rate of free radical formation increases, which is in accordance with Fenton-like previous studies [34].

### 3.3. The Possible Mechanism of Terramycin Degradation

#### 3.3.1. Identification of Mainly Free Radical Species

According to previous literature, •OH and •O_2_^−^ can be generated from the activation of an iron catalyst for H_2_O_2_ in a Fenton-like system for the degradation of organic pollutants under acidic or neutral conditions [35,36]. To identify the free radical species formed in the present system, an EPR experiment was also carried out, and the EPR spectra are found in Figure 7a,b, which displays a strong 4-fold characteristic peak of the typical DMPO-•OH adduct with an intensity ratio of 1:2:2:1, and a relatively weak 1:1:1:1 quadruplet peak of DMPO-•O_2_^−^ adducts was also discovered at the same time. Moreover, the intensities of these peaks increase with prolonged degradation times, demonstrating that •OH and •O_2_^−^ are important active species in degradation processes. According to literature reports [21,22], HO_2_• plays an important role in the Fenton-like reaction; however, no peak about HO_2_• is observed in the EPR spectrum, which may be because of the rapid consumption of the HO_2_• generated during the degradation reaction, resulting in no HO_2_• to react with DMPO and produce a signal.

#### 3.3.2. Catalytic Degradation Pathway

Based on all the results obtained above and discussed below, a possible mechanism for catalytic degradation was proposed in Figure 8. FTIR and XPS analyses illustrate that rGO-coated CuFeO_2_ had a large number of groups on its surface, which enhanced the adsorption of terramycin by rGO-coated CuFeO_2_. At first, terramycin is adsorbed with -OH of rGO through a hydrogen bond, while the aromatic ring of rGO is bonded to terramycin through the π-π stack. Then, a series of chemical reactions take place and are expressed in Equations (3–8). Equation (3) shows Fe(III) is reduced by Cu(I) to produce Fe(II), which activates H_2_O_2_ to generate •OH according to the Haber–Weiss mechanism in Equation (5). Simultaneously, regeneration of Fe(II) from Fe(III) could be achieved using Equation (4). Similar to Fe(II), Cu(I), can activate H_2_O_2_ to generate •OH, and regeneration of Cu(I) could be achieved from Cu(II) in Equations (6) and (7), respectively. In addition, O_2_ gains electrons to form •O_2_^−^ in Equation (8).
Fe(III) + Cu(I) Fe(II) + Cu(II)(3)
Fe(III) + H_2_O_2_ Fe(II) + HO_2_• + H^+^(4)
Fe(II) + H_2_O_2_ Fe(III) + HO• + OH^-^(5)
Cu(I) + H_2_O_2_ Cu(II) + HO• + OH^-^(6)
Cu(II) + H_2_O_2_ Cu(I) + HO_2_• + H^+^(7)
O_2_ + e^−1^ •O_2_^−^(8)

## 4. Conclusions

A novel Fenton-like rGO-coated CuFeO_2_ catalyst was successfully synthesized by a hydrothermal reaction procedure with a micro- and mesoporous structure. The test results from XPS and FTIR indicated that there were a large number of groups, including C=C/C−C, CH_2_−CO, and HO−C=O, on the surface of the catalyst, which was conducive to the adsorption of terramycin. The catalytic degradation of terramycin by rGO-coated CuFeO_2_ can be described by a pseudo-second-order kinetic model, and the degradation reaction is more intense and complete at higher temperatures. EPR spectra revealed that •OH and •O_2_^−^ are important active species in degradation processes. In the catalytic degradation process of terramycin, Fe(III) and Cu(II) were continuously reduced to produce Fe(II) and Cu(I), ensuring a continuous degradation reaction and thus providing an economical and environmentally friendly antibiotic degradation technology.

## Figures and Tables

**Figure 1 nanomaterials-12-04391-f001:**
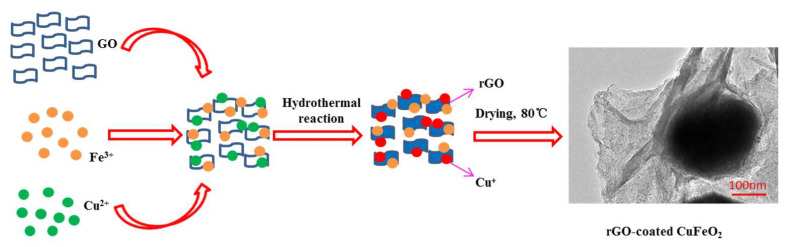
The formation mechanism of the rGO-coated CuFeO_2_.

**Figure 2 nanomaterials-12-04391-f002:**
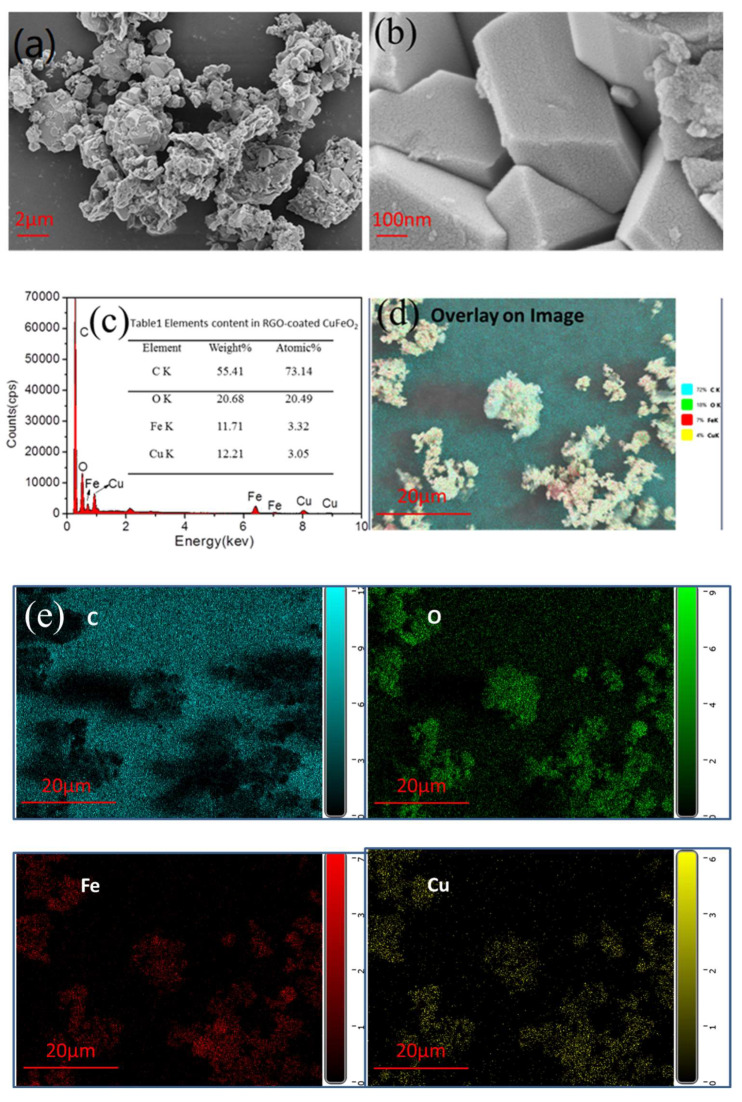
(**a**,**b**) SEM images; (**c**) EDX spectra; (**d**) the real-time surface distribution map; and (**e**) the elemental mapping images.

**Figure 3 nanomaterials-12-04391-f003:**
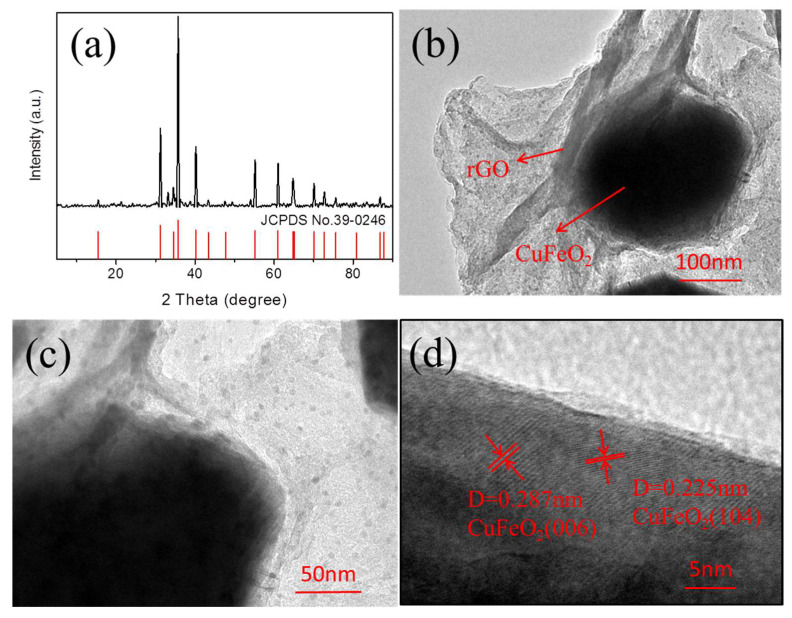
(**a**) XRD; (**b**,**c**) TEM images; and (**d**) HRTEM.

**Figure 4 nanomaterials-12-04391-f004:**
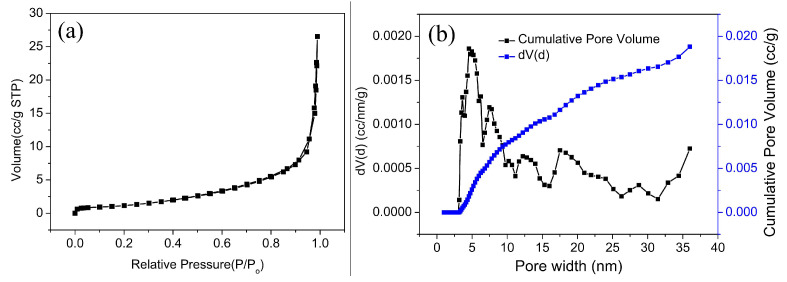
(**a**) Nitrogen adsorption-desorption isotherm and (**b**) pore size distribution.

**Figure 5 nanomaterials-12-04391-f005:**
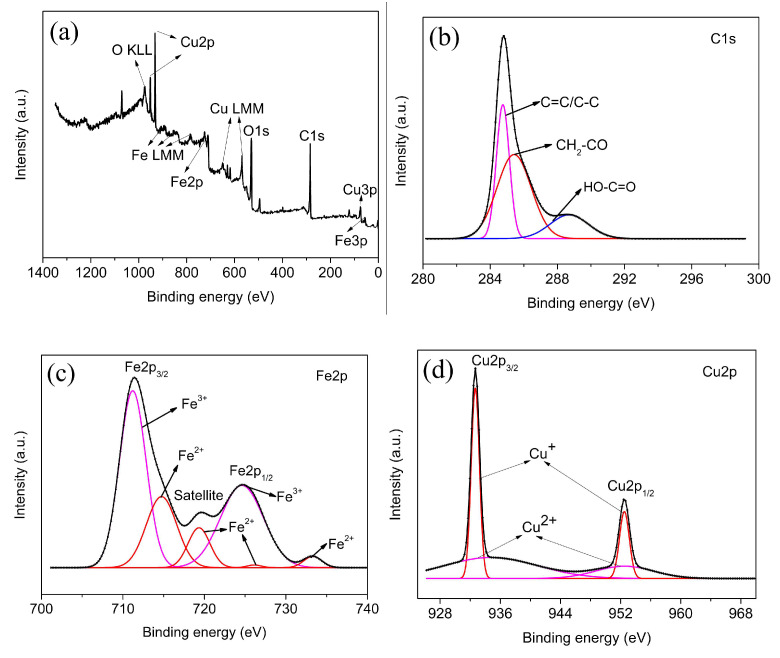
(**a**) XPS spectra; (**b**) C 1s spectra; (**c**) Fe 2p spectra; (**d**) Cu 2p spectra; (**e**) O 1s spectra; and (**f**) FTIR.

**Figure 6 nanomaterials-12-04391-f006:**
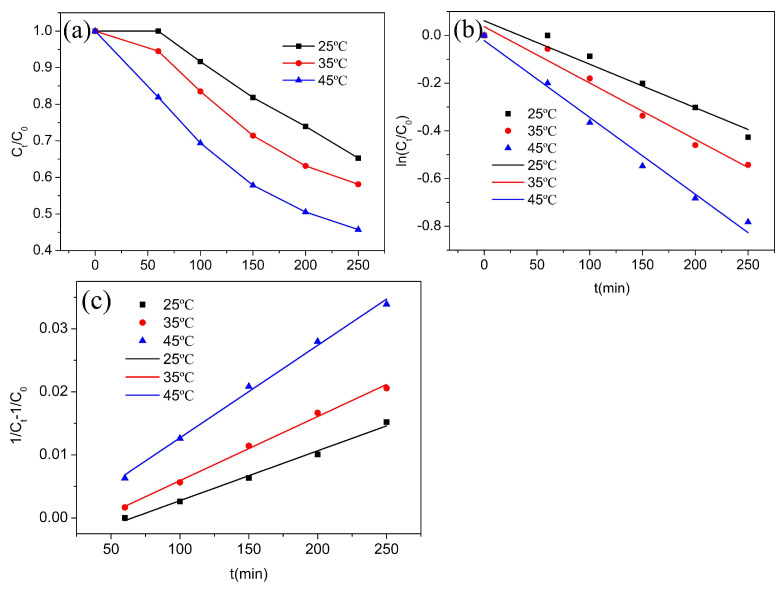
(**a**) Removal of terramycin in the rGO-coated CuFeO_2_/H_2_O_2_ system at varying temperatures; (**b**) pseudo-first-order kinetics; and (**c**) pseudo-second-order kinetics.

**Figure 7 nanomaterials-12-04391-f007:**
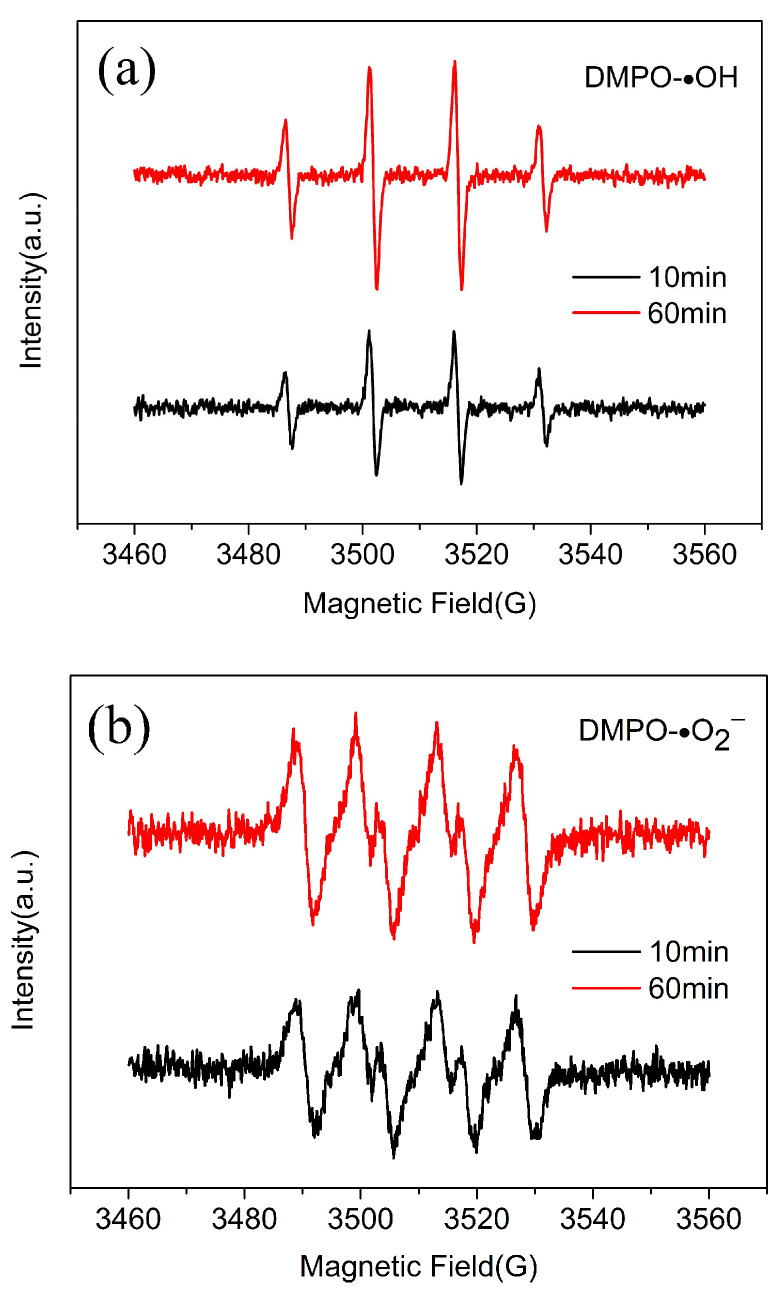
EPR spectra of (**a**) DMPO-•OH adducts and (**b**) DMPO-•O_2_^−^ adducts.

**Figure 8 nanomaterials-12-04391-f008:**
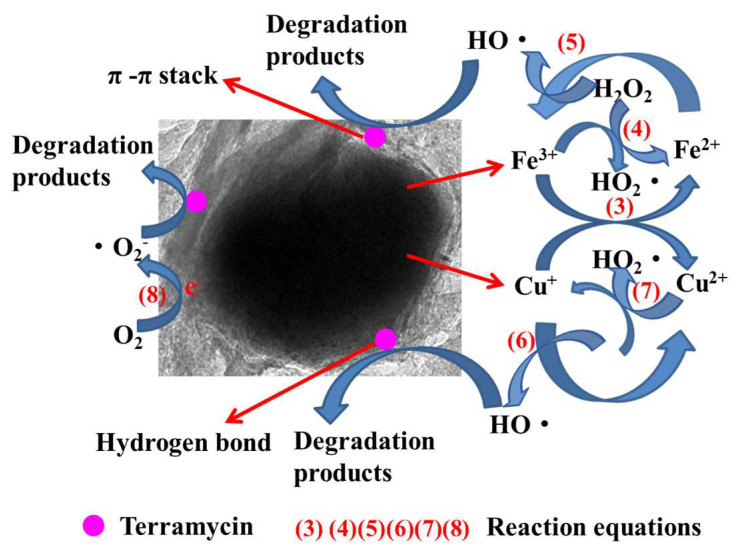
Degradation mechanism diagram.

**Table 1 nanomaterials-12-04391-t001:** Element content of RGO-coated CuFeO_2_.

Element	Weight%	Atomic%
C K	55.41	73.14
O K	20.68	20.49
Fe K	11.71	3.32
Cu K	12.21	3.05

**Table 2 nanomaterials-12-04391-t002:** The pseudo-first-order kinetics and pseudo-second-order kinetics fitting parameters.

Kinetics Models	Parameters	25 °C	35 °C	45 °C
Pseudo-first-order kinetics	k_1_ (min^−1^)	0.00182	0.00236	0.00322
	r^2^	0.9296	0.9733	0.9851
Pseudo-second-order kinetics	k_2_ (L·mg^−1^·min^−1^)	7.907 × 10^−5^	1.018 × 10^−4^	1.467 × 10^−4^
	r^2^	0.9907	0.9945	0.9948

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
