# Peer review of "Reduced Graphene Oxide-Coated CuFeO2 with Fenton-like Catalytic Degradation Performance for Terramycin"

_nanomaterials, 2022, doi:10.3390/nano12244391_

Round 1
Reviewer 1 Report
In this article the author reports ‘Reduced Graphene Oxide-coated CuFeO2 with Fenton-like 2
Catalytic Degradation Performance for Terramycin’. This topic will be surely interesting for many researchers working in the related fields. However, there are some points that need to be addressed before possible publication.
Recommendation: Major as noted
1. The novelty of the present article should be discussed a little more in the Introduction section.
2. Scale bars in Fig.2e and Fig.3b,c,d are not clearly visible to the readers.
3. The author should write the purpose for each test in one/two sentences (in brief) before explaining the results of the characterization techniques. Therefore, the logic and organization of this part will be enhanced.
4. The formatting and grammatical errors in the article need to be checked carefully.
5. The Figure captions of Fig.3 should be improved with a clear indication of the material.
6. The authors have cited relevant references in the Introduction section; however the manuscript needs to be highlighted with some recent reports to further broaden the impact, related literatures: Chemical Engineering Journal, 433, 134609; The Journal of Physical Chemistry B, 122(29), 7201-7218; Environmental Pollution, 244, 93-101; Polymers, 12(3), 555; RSC Advances, 5(21), 15861-15869; ACS Applied Nano Materials, 5(1), 1644-1655.
7. Conclusion should be supported by important results of the present work.
Author Response
Dear reviewer,
We are truly grateful to your critical comments and thoughtful suggestions. Based on these comments and suggestions, we have made careful modifications. We hope the revised manuscript will meet the magazine’s standard. Below you will find our point-by-point responses to your question:
- The novelty of the present article should be discussed a little more in the Introduction section.
Response: The novelty of the present article has been discussed in the Introduction section.
- Scale bars in Fig.2e and Fig.3b,c,d are not clearly visible to the readers.
Response: Fig.2e and Fig.3b,c,d have been modified.
- The author should write the purpose for each test in one/two sentences (in brief) before explaining the results of the characterization techniques. Therefore, the logic and organization of this part will be enhanced.
Response: This part has been improved.
- The formatting and grammatical errors in the article need to be checked carefully.
Response: The formatting and grammatical errors in the article have been checked carefully.
- The Figure captions of Fig.3 should be improved with a clear indication of the material.
Response: The Figure captions of Fig.3 has been improved.
- The authors have cited relevant references in the Introduction section; however the manuscript needs to be highlighted with some recent reports to further broaden the impact, related literatures: Chemical Engineering Journal, 433, 134609; The Journal of Physical Chemistry B, 122(29), 7201-7218; Environmental Pollution, 244, 93-101; Polymers, 12(3), 555; RSC Advances, 5(21), 15861-15869; ACS Applied Nano Materials, 5(1), 1644-1655.
Response: Relevant references have been added to the paper.
- Conclusion should be supported by important results of the present work.
Response: The conclusion has been modified accordingly.
Thanks
Sincerely yours,
Liping Wang, On behalf of all authors.

Reviewer 2 Report
This work reports the hydrothermal synthesis and physicochemical characteristics of reduced graphene oxide-coated CuFeO2 with its catalytic performance in terramycin degradation. Accordingly, the authors suggest the degradation mechanism on the catalyst. This work fits the aims and scope of ‘Nanomaterials’ and is adequate to its wide readership. The research is well accomplished, but the introduction, results and discussion can be improved before the manuscript can be accepted.
1. In the section of the Introduction, it is suggested that the Fenton process should be expressed in details.
2. The ‘freeze drying’ of Figure 1 does not appear in the text of the catalyst preparation, which should be corrected relevantly.
3. In line 110, the expression of ‘suggesting the generating of single-phase CuFeO2’ is not appropriate in the results of EDX analysis, as it could be suggested by XRD rather than by EDX.
4. The CuFeO2 crystalline of the HRTEM in Figure 3d seems to be present in the shell side. The coating of rGO on the CuFeO2 particles is needed to be explained.
5. In XPS C1s spectrum, the presence of HO-C=O is observed. In addition, the hydroxyl groups are found in XPS O 1s and FTIR. However, the existence of C=O is not displayed in the related spectra, of which a reason is necessary to be proposed.
6. The Ct/C0 deceases from 0.653 to 0.457 with increasing the reaction temperature. Please, confirm the expression in the line 178.
7. Please, unify the expression in ‘pseudo-‘ and ‘quasi-‘ of the model. In the pseudo(quasi)-second-order model, the unit of k2 parameter should be corrected to Lmg-1min-1.
8. It is suggested that the section of ‘catalytic degradation pathway’ should be improved to clearly explain the roles of rGO and CuFeO2. Moreover, the equations from (3) to (8) need to be included in the explanation. Even though the pink circles in Figure 8 might be expected to terramycin, please clarify the notification in the diagram. It could be helpful to match the arrows in Figure 8 to the reaction equation numbers.
Author Response
Dear reviewer,
We are truly grateful to your critical comments and thoughtful suggestions. Based on these comments and suggestions, we have made careful modifications. We hope the revised manuscript will meet the magazine’s standard. Below you will find our point-by-point responses to your question:
- In the section of the Introduction, it is suggested that the Fenton process should be expressed in details.
Response: The Fenton process has been expressed in details.
- The ‘freeze drying’ of Figure 1 does not appear in the text of the catalyst preparation, which should be corrected relevantly.
Response: Figure 1 has been corrected.
- In line 110, the expression of ‘suggesting the generating of single-phase CuFeO2’ is not appropriate in the results of EDX analysis, as it could be suggested by XRD rather than by EDX.
Response: The results of EDX analysis has been modified.
- The CuFeO2 crystalline of the HRTEM in Figure 3d seems to be present in the shell side. The coating of rGO on the CuFeO2 particles is needed to be explained.
Response: The coating of rGO on the CuFeO2 particles can be seen in Fig. b and c. The corresponding part has been modified.
- In XPS C1s spectrum, the presence of HO-C=O is observed. In addition, the hydroxyl groups are found in XPS O 1s and FTIR. However, the existence of C=O is not displayed in the related spectra, of which a reason is necessary to be proposed.
Response: The corresponding content has been modified. According to reference 22, the peak at 533.1 eV attributes to the oxygen of physically absorbed H2O molecules (H2O)/C=O in the high-resolution spectrum of O 1s. In addition, the absence of the C=O peak in HO-C=O may be due to hydrogen bonding between C=O and -OH which reduces vibration frequency of C=O.
- The Ct/C0 deceases from 0.653 to 0.457 with increasing the reaction temperature. Please, confirm the expression in the line 178.
Response: This part has been modified as “The Ct/C0 is 0.653 at 25 ℃ and decreases to 0.457 at 45 ℃ when the catalytic degradation time is 250min”.
- Please, unify the expression in ‘pseudo-‘ and ‘quasi-‘ of the model. In the pseudo(quasi)-second-order model, the unit of k2 parameter should be corrected to Lmg-1min-1.
Response: The expression of the kinetics models has been unified using “pseudo-”. In the pseudo-second-order model, the unit of k2 parameter has been corrected to L•mg-1•min-1.
- It is suggested that the section of ‘catalytic degradation pathway’ should be improved to clearly explain the roles of rGO and CuFeO2. Moreover, the equations from (3) to (8) need to be included in the explanation. Even though the pink circles in Figure 8 might be expected to terramycin, please clarify the notification in the diagram. It could be helpful to match the arrows in Figure 8 to the reaction equation numbers.
Response: The section of ‘catalytic degradation pathway’ has been improved and the equations from (3) to (8) have been included in the explanation. Moreover, the notification in the diagram has been clarified in Figure 8.
Thanks
Sincerely yours,
Liping Wang, On behalf of all authors.

Round 2
Reviewer 1 Report
The authors have addressed all the questions raised before, therefore the manuscript can be accepted in the present form.